# Tracking Epidermal Cortisol and Oxytocin in Managed Bottlenose Dolphins as Potential Non-Invasive Physiological Welfare Indicators

**DOI:** 10.3390/ani15172628

**Published:** 2025-09-08

**Authors:** Clara Agustí, Oriol Talló-Parra, Enrique Tejero-Caballo, Daniel Garcia-Parraga, Marina López-Arjona, Teresa Álvaro-Álvarez, José Joaquín-Cerón, Xavier Manteca

**Affiliations:** 1Department of Animal and Food Science, School of Veterinary Science, Universitat Autònoma de Barcelona, 08193 Bellaterra, Spain; clara.agustipuj@autonoma.cat (C.A.); marina.lopez10@um.es (M.L.-A.); xavier.manteca@uab.cat (X.M.); 2Animal Welfare Education Centre (AWEC), School of Veterinary Science, Universitat Autònoma de Barcelona, 08193 Bellaterra, Spain; 3Research Department, Fundación Oceanogràfic de la Comunitat Valenciana, Ciudad de las Artes y las Ciencias, 46013 Valencia, Spain; enriquetejero16@usal.es (E.T.-C.); dgarcia@oceanografic.org (D.G.-P.); 4Veterinary Services, Oceanogràfic, Ciudad de las Artes y las Ciencias, 46013 Valencia, Spain; talvaro@oceanografic.org; 5Interdisciplinary Laboratory of Clinical Analysis, Interlab-UMU, Regional Campus of International Excellence Campus Mare Nostrum, University of Murcia, 30100 Murcia, Spain; jjceron@um.es

**Keywords:** cetaceans, welfare assessment, welfare indicators, non-invasive methods, wellbeing, odontocetes, bottlenose dolphin, cortisol, oxytocin, epidermis

## Abstract

Public concern about the welfare of dolphins and other cetaceans, both in human care and in the wild, is increasing. Yet, assessing their wellbeing remains a challenge, especially through non-invasive and scientifically robust methods. This study explored the potential of two hormones naturally present in dolphin skin—cortisol, related to stress, and oxytocin, linked to social bonding and wellbeing—as physiological indicators of welfare. Using skin samples from five bottlenose dolphins monitored over several months, we validated laboratory methods for measuring these hormones and examined how their levels varied in response to real-life events. Epidermal cortisol levels were lower after mild weight loss and diazepam administration and varied seasonally. Oxytocin levels tended to decrease during periods with more negative welfare indicators and during park closures related to the COVID-19 pandemic and tended to increase with higher visitor presence and diazepam administration. These results suggest that both hormones reflect relevant changes in the animals’ environment and health, although interpreting oxytocin remains challenging. Our findings highlight the promise of skin-based hormonal markers to monitor dolphin welfare over time. With further validation, this approach could improve how we assess and promote cetacean wellbeing, contributing to better care, management, and conservation strategies.

## 1. Introduction

In recent years, growing public concern over emerging threats to cetacean welfare, both in captivity and in the wild [1,2], has driven welfare considerations to shape facility policies and influence national and international regulations on cetacean protection [3]. In light of these pressures, establishing robust, scientifically rigorous frameworks for assessing and improving their welfare is more critical than ever [4,5]. Building on global advancements in animal welfare science, the field is moving beyond theoretical or anecdotal perspectives toward empirical research that closely examines the specific needs and affective states of cetaceans in diverse environments [6,7,8]. Consequently, modern accredited zoological institutions and other key stakeholders are increasingly employing valid welfare measures to guide ethical decision-making and inform policy development [9,10]. Nevertheless, conditions in managed care facilities remain a significant concern, as critical substantial knowledge gaps persist in defining, assessing, and maximizing cetacean welfare.

Animal welfare is a complex science that requires diverse approaches for accurate assessments. Among these, physiological indicators serve as valuable “nonspecific” measures of welfare changes [11,12]. Exposure to stressful stimuli usually activates the hypothalamic–pituitary–adrenal (HPA) axis, leading to increased secretion of glucocorticoids from the adrenal cortex, which play a primary role in mediating physiological responses to both acute and chronic stress [13]. Cortisol, the primary glucocorticoid secreted in marine mammals, serves a critical adaptive function by mobilizing energy during acute challenges and by modulating metabolism, immunity, cardiovascular tone, and reproduction [14,15]. However, chronic activation of the HPA axis may result in adverse effects such as immunosuppression, impaired reproduction, and reduced growth [11].

Meanwhile, attention in animal welfare science has increasingly turned to markers that can capture positive emotions, particularly oxytocin [16]. Oxytocin, mainly produced in the hypothalamus and released by the posterior pituitary [17], is involved not only in reproduction and lactation [18] but also in social bonding, cognition, and positive affective states [19,20], while buffering stress [16,21]. In cetaceans, where important social relationships extend beyond mating and parental care [22,23], understanding oxytocin’s role may offer valuable insights into welfare, mirroring insights from primate research [24].

In cetaceans, cortisol has been quantified in several matrices (blood, respiratory droplets, blubber) and, more recently, in the epidermis and baleen, each providing information over different temporal windows of HPA-axis activity. Mid-term fluctuations are often captured in blubber or feces [25,26], whereas keratinous tissues such as the epidermis and baleen integrate hormone deposition over longer periods [27,28]. Cortisol is thought to diffuse from blood into developing skin cells during keratinization [29], and epidermal concentrations have been linked to welfare-related changes in managed bottlenose dolphins [30] and belugas [31], as well as in stranded striped dolphins [32]. Additionally, local synthesis in the skin itself [33] suggests another potential pathway for cortisol accumulation in the epidermis.

By contrast, oxytocin data are scarce and mostly limited to blood samples from free-ranging cetaceans [34]. Because oxytocin is a large, lipid-insoluble peptide, passive diffusion from circulation into the epidermis is less probable [35], making active transport a more plausible route [36]. Nevertheless, like cortisol, oxytocin can be synthesized locally by keratinocytes [37], and keratinized tissues may still serve as reservoirs [27,38].

Building on these findings, monitoring epidermal cortisol concentrations (ECCs) and epidermal oxytocin concentrations (EOCs) may offer a non-invasive method for assessing cetacean welfare and understanding how multiple stressors impact it. However, for each species, it must be first ensured that the methods reliably distinguish different hormone concentrations and then verify that these measurements accurately reflect the animals’ physiological status [39]. Moreover, to reliably infer the welfare of individual animals, any indicator must be validated to confirm it measures the intended affective states, demonstrating a direct relationship, whether causal or consequential, with those states [40]. Structured frameworks to evaluate animal welfare, such as the Five Domains Model, can support this validation process by integrating welfare indicators into four physical/functional domains (nutrition, environment, health, and behavior) and a fifth domain focused on the mental state [41,42].

In wild species, validating new physiological welfare indicators requires studying populations with well-documented welfare and establishing normal ranges by age, sex, reproductive condition, and possibly season or diet [39,43,44]. Once validated, these measures can be extended to other less known populations (e.g., beluga whales [45,46]). Because researching wild populations is inherently challenging, many studies developing new biomarkers for cetaceans rely on individuals in managed care, where frequent, minimally stressful sampling, detailed health records, and systematic monitoring are feasible [47,48].

In a broader effort to advance the validation of novel physiological indicators for monitoring cetacean welfare, this study pursued the following objectives: (i) to develop and validate a method for extracting and quantifying cortisol and oxytocin from the epidermis of common bottlenose dolphins (*Tursiops truncatus*) using AlphaLISA immunoassays; (ii) to investigate the potential influence of environmental and welfare-related factors on the ECC and EOC, using routine weekly samples collected over an extended period of time from five captive individuals; (iii) to compare the ECC and EOC in two individuals exposed to a significant stressor (inter-facility transport) against values observed in the long-term monitored cohort; (iv) to examine correlations between the ECC and EOC; and (v) to explore the potential of the ECC and EOC as indicators of long-term welfare in cetaceans, considering possible time lags between HPA axis and oxytocin system activity and hormone detectability in the skin.

## 2. Materials and Methods

### 2.1. Ethics Statement

All procedures were conducted under a protocol approved by the Institutional Animal Care and Welfare Committee of the Oceanogràfic (Project reference: OCE-33-18/3 August 2018), which is authorized by the Spanish regional government as an enabled body for the evaluation of research projects with live animals (ID ES460250001014). Skin sampling in this study was facilitated by positive reinforcement training, ensuring that procedures did not cause injury or disrupt the dolphins’ usual routines, and took place only under voluntary participation; otherwise, it was postponed. Oceanogràfic de València aquarium is accredited by organizations such as the European Association for Aquatic Mammals (EAAM), European Association of Zoos and Aquaria (EAZA), and the American Zoo and Aquarium Association (AZA), being a Global Humane Certified Aquarium ensuring adherence to established animal care and higher welfare standards.

### 2.2. Study Area and Individuals

Five bottlenose dolphins housed at the Oceanogràfic de Valencia aquarium (City of Arts and Sciences, Comunidad Valenciana, Spain) were included in this study. These individuals were selected from a larger population of 18 dolphins and comprised three males, aged 7 (M1), 14 (M2), and 16 (M3) years, and two females, aged 16 (F1) and 35 (F2) years. Selection was non-random and specifically targeted individuals who, based on routine welfare monitoring, exhibited a higher frequency of negative welfare indicators than the other individuals housed in the same facility during the study period (see Section 2.6). As such, the sample is not representative of the entire population at the facility but was selected to enable the assessment of intra-individual changes in hormonal and welfare-related variables in response to fluctuating welfare conditions. Two additional females, aged 15 (F3) and 18 (F4) years, were transferred from Parc Astérix (Plailly, France) on 24 January 2021 and were included in the study as an opportunity to assess epidermal biomarkers during a potentially stressful context involving transport and acclimation.

The dolphin facility comprises seven outdoor pools up to 11 m deep, interconnected by gates that can be opened or closed as needed, with a total capacity of 23 million liters of water. Water temperatures are controlled ranging from 19.2 to 26.3 °C, maintaining the animals within their thermal neutral zone [49,50]. All animals receive a balanced diet of frozen fish, including herring (*Clupea harengus*), capelin (*Mallotus villosus*), hake (*Merluccius merluccius*), and squid (*Loligo* spp.), formulated to meet individual requirements. Positive reinforcement training is the primary strategy used for husbandry, veterinary, and research activities.

### 2.3. Sample Collection and Preparation

A single scraped epidermis sample was routinely collected from each individual every Friday between January 2019 and October 2020. However, sampling was not feasible every week for all individuals due to the animals’ voluntary participation, trainer availability, and the need to obtain a sufficient sample mass (M1: N = 30; M2: N = 32; M3: N = 55; F1: N = 33; F2: N = 45). Additionally, handling and epidermal sampling of F3 and F4, transported on 24 January 2021, was not possible until 15 March (F3), 21 March (F3 and F4), and 4 April (F3), approximately 50–70 days after their arrival due to the acclimation period. However, they were included in the study given their potential to provide insight into physiological changes associated with the post-transport adjustment phase.

Epidermal sampling involved positioning the dolphin parallel to the pool edge with the target area out of the water, followed by drying the area with a gauze pad. A semi-rigid plastic card (smooth-edged and disinfected with alcohol) was used to gently scrape desquamated epidermis (≈15 cm × 15 cm) from the dorsal fin, identified as an optimal site for animal and trainer comfort, applying moderate pressure in multiple directions. Three to six scrapings were transferred into a 1.5 mL Eppendorf tube and stored at −20 °C (see Agustí et al. [27] for more details).

### 2.4. Sample Storage, Preparation, and Hormone Extraction

Frozen skin samples were first dried in an oven (Heraeus model T6; Kendro^®^ Laboratory Products, Langenselbold, Germany) at 36 °C for 72 h to evaporate any remaining water. To account for the effect of the sample mass previously reported [27], samples were normalized to 15–20 mg, and those weighing less than 15 mg were excluded from the analysis. Each sample was then pulverized to a fine powder using a Precellys Evolution homogenizer (Bertin Technologies, Montigny-le-Bretonneux, France). The pulverized epidermis was incubated in 1 mL of methanol for 18 h at room temperature under continuous, gentle agitation, ensuring that no skin particles adhered to the tube caps. After incubation, the samples were centrifuged at 16,000× *g* for 10 min, and 0.6 mL of the supernatant was transferred to a new tube. This supernatant was evaporated to dryness over 2 h in a Speed Vac Concentrator (Concentrator 5301, Eppendorf, Hamburg, Germany), and the resulting dry extract was reconstituted in 100 µL of phosphate-buffered saline (PBS) for further analysis.

### 2.5. Hormone Detection and Assay Validation

Oxytocin and cortisol were quantified using AlphaLISA^®^ immunoassays (PerkinElmer, Shelton, CT, USA). The oxytocin assay was run in a direct competitive format, whereas the cortisol assay followed an indirect competitive format. Both assays employ two types of beads (donor and acceptor) and rely on the competition between endogenous and biotinylated hormone for a monoclonal antibody specific to the target analyte, previously validated for use in sow hair (cortisol [51]; oxytocin: [38]). In our validations, cross-reactivity was assessed only against cortisone for the cortisol antibody (1.03%) [51] and against vasopressin for the oxytocin antibody, with no detectable cross-reactivity observed [38]. Other structurally related compounds were not tested. All laboratory extraction and analysis procedures were performed by the same two trained researchers to minimize inter-operator variability.

In line with Reimers and Lamb’s [52] validation criteria, the accuracy, precision, specificity, and sensitivity of the cortisol and oxytocin AlphaLISA assays were evaluated using pooled epidermal samples. Standards were prepared by diluting conjugated oxytocin (Oxytocin–BSA) and cortisol (cortisol-KLH, Cloud-Clone, Katy, TX, USA) in AlphaLISA Universal buffer, generating a standard curve with eight concentrations. Specificity was assessed through dilution linearity, by serially diluting two epidermal extracts from 1:2 to 1:128 in AlphaLISA Universal buffer and verifying that signal responses remained proportional to the dilution, indicating minimal matrix interference. Accuracy was evaluated via spike-and-recovery tests. Known concentrations of conjugated oxytocin and cortisol were added to epidermal extracts with previously quantified hormone levels. Recovery percentages were calculated by comparing measured versus expected concentrations. Precision was determined by calculating intra- and inter-assay coefficients of variation (CVs). Intra-assay precision was assessed by analyzing five replicates of two sample pools (high and low hormone concentrations) within a single run. Inter-assay precision was evaluated by measuring these samples in duplicate across multiple days, using freshly prepared calibration curves each time. Sensitivity was determined by identifying the lowest hormone concentration that could be reliably detected by the assay.

### 2.6. Data Collection and Individual Welfare Profiling Based on Aquarium Reports

In parallel with epidermis sampling, a comprehensive list of welfare indicators was developed following the Five Domains Model framework [42]. This list was specifically tailored to the context of this study, drawing on established frameworks for bottlenose dolphins such as C-Well [53] and Dolphin WET [54] and aligned with similar approaches applied to other species (e.g., [55,56]). It incorporates recorded welfare indicators alongside their inferred affective states (Domain 5), with the aim of providing a structured framework for assessing welfare in this study (Table 1).

The indicators were identified using the aquarium’s existing welfare-monitoring strategies, which are an integral part of the daily care routine for Oceanogràfic’s dolphins. Aquarium staff regularly record various aspects of dolphin welfare through reports and observations, primarily to promptly detect health or behavioral issues rather than to quantify behavior frequency. These records span both training sessions and free time periods (e.g., free-swim time, social interactions, or enrichment activities), as well as daily interactions with veterinarians (e.g., medical checks) or researchers. Welfare-related information from trainers’ daily reports and veterinary records (Table 1) was systematically converted into welfare indicators, which were scored daily in a binary format as either absent (0) or present (1).

Importantly, certain negative welfare indicators described by EAAM [50] and Baumgartner et al. [54] were monitored throughout the study but were never observed. These included water-quality or temperature parameters falling outside the safe and comfortable ranges, poor food quality, inadequate hydration, weight fluctuations exceeding 13% of body weight over the year, trainer-directed aggression, incidents or diseases affecting locomotion or buoyancy, prolonged floating, apathy, anorexia, and severe interspecific aggression leading to serious injury or wounds. Consequently, these indicators did not appear in the data and were excluded from the final welfare-indicator list despite being monitored.

The administration of diazepam at anxiolytic doses was also included in the study due to its known effects on the HPA axis and behavior. Specifically, diazepam was occasionally administered to a few individuals to promote social stability during conflicts, mitigate suspected stress-related gastric ulcers, and facilitate acclimation to environmental and social change [68], in order to prevent situations that might otherwise escalate into more serious conflicts and compromise welfare [58]. This primarily involved three male individuals, specifically selected for the study due to their subordinate status and heightened susceptibility to social stress arising from conflicts with dominant male conspecifics. Information on diazepam administration was obtained from veterinary records and scored daily in a binary format as either absent (0) or present (1). Additionally, we considered environmental and contextual factors that could potentially influence hormone levels, including the season and visitor-based seasonality. The season was determined based on calendar divisions: Spring (21 March–20 June), Summer (21 June–20 September), Fall (21 September–20 December), and Winter (21 December–20 March). Subsequently, water temperature records were classified by season, and distributions were compared.

Visitor-based seasonality was determined based on attendance patterns at the aquarium: the Peak season (1 July–31 August 2019) corresponded to the months with the highest visitor attendance, during which the aquarium operated for extended hours and scheduled 4 to 6 daily educational presentations involving different dolphins. The Off-peak season (1 January–30 June 2019; 1 September 2019–11 March 2020; 1 July–30 October 2020) encompassed periods of generally lower visitor numbers, although they fluctuated depending on non-working days and special events, with only 2 to 4 daily educational presentations with dolphins scheduled. Finally, a third period named the Closure period (12 March–30 June 2020) was included. It corresponded to the temporary shutdown due to COVID-19 lockdown measures, during which the aquarium remained closed to the public.

### 2.7. Data Management

Due to the large number of recorded variables, data management aimed at simplifying the data structure was recommended to reduce model complexity while preserving variables relevant to animal welfare and physiology. Most welfare indicators occurred infrequently (<5% of days), potentially introducing noise and limiting the ability to draw meaningful conclusions. To address this issue, all welfare indicators assumed to negatively impact the welfare state (Table 1) were combined into a single cumulative variable, termed ‘Negative welfare indicators’. Since the variable Negative welfare indicators was composed of the sum of 12 variables, its daily range spans from 0 to 12.

The variable Willingness to participate in training sessions, was excluded from all the statistical analyses due to incomplete and inconsistent data recording during the Closure period, as well as limited daily variability (scores were frequently recorded at the value of 3/4, indicating generally high motivation during training sessions). These issues introduced potential bias and reduced the interpretability of the variable across the full study period. Nevertheless, descriptive data for this variable were retained and used in the discussion, as they provided valuable contextual and welfare-related information that supported the interpretation of the other findings.

To include season categories in the model, we created three dummy variables: Spring, Summer, and Fall. These variables took a value of 1 when observations fell within their respective season and 0 otherwise, while Winter was deliberately left out as the baseline category. This approach conditioned the model to compare each non-baseline category against Winter while preventing collinearity. Similarly, Visitor-based seasonality was represented with dummy variables for Peak season and Closure, with Off-peak season serving as the baseline.

Then, all daily data were aggregated into weekly spans (Monday to Sunday) by summing daily scores to obtain a single value per week (Table 2). The classification of each week into both a season and a Visitor-based seasonality category was based on date ranges: weeks lying wholly within a period were assigned to that period, and weeks spanning a transition were assigned to the period for which the start date fell within the week. Overall, weekly aggregation provided a more representative framework for capturing the potential integrative nature of the matrix and allowed single-point weekly hormone measurements to be directly associated with weekly integrated welfare indicators. Because the Negative welfare indicators variable was derived by summing 12 daily measures, resulting in a different scale from the other variables, all predictors were standardized prior to the statistical analysis.

To account for the lag between actual HPA axis activity and the corresponding hormonal signal potentially detectable in the collected epidermis, hormone data were shifted backward in time by the appropriate lag duration. This adjustment assumed a linear relationship between the weekly integrated data (predictor variables) and hormone levels. Analyses were repeated for each of the following time lags to capture a range of potential offsets between the central and local HPA axis or oxytocin system activity and epidermal hormone detection: 20–26, 27–33, 34–40, 41–47, 48–54, 55–61, 62–68, and 69–75 days. Given the current lack of precise knowledge on the exact time lag between endocrine events and their epidermal reflection, this range of plausible lag intervals was tested to ensure that potential effects were not overlooked. The selection of tested time lags was based on studies determining epidermal turnover in one bottlenose dolphin [69], the suggested time lag between peak blood cortisol and its detection in epidermal tissue in two bottlenose dolphins [30], and the proposed mechanisms of hormone accumulation and/or synthesis in the epidermis for cortisol [29,70] and oxytocin [37,71].

### 2.8. Statistical Analyses

All analyses were conducted in R (version 4.3.3; [72]), with a significance threshold set at *p* < 0.05. All values are reported as the mean ± standard deviation (SD). A Shapiro–Wilk test was performed prior to analysis to assess the normality of all variables, using the ‘shapiro.test’ function from the ‘stats’ package.

Water temperature was classified by season, and differences in rank distributions were assessed using a Kruskal–Wallis (KW) test due to non-normality, implemented via the ‘kruskal.test’ function from the ‘stats’ package. Post-hoc pairwise Wilcoxon rank-sum tests with Bonferroni correction were conducted to identify significant differences between season categories.

Then, separate models were conducted to study the relationships between predictors related to the environment and animal welfare and ECC and, independently, EOC. Additionally, each hormone model was tested separately for the previously described time lags. Linear models (LMs) were applied to all analyses using the ‘lm’ function from the ‘stats’ package, as they offered a stable and interpretable analytical framework given the structure and size of the dataset. Although LMs do not explicitly account for inter-individual variability through random effects, they are generally robust to moderate deviations from normality and were considered appropriate for the exploratory nature of this study.

We fitted an LM of the following form:Y = β0 + β1X1 +…+ βkXk + ε,
where Y represents the response variable (ECC or EOC), X1, …, Xk are predictor variables, and ε is the error term, assumed to be normally distributed with mean zero and constant variance. Coefficients (β) were estimated using ordinary least squares (OLS), and predictor significance was evaluated based on standard statistical criteria. Key model assumptions, including the normality of residuals, homogeneity of variance (homoscedasticity), and absence of multicollinearity, were carefully assessed. While residuals showed deviations from normality and homoscedasticity, LMs are generally robust to moderate deviations from normality due to the Central Limit Theorem. However, significant departures from homoscedasticity can affect the reliability of estimates and their interpretation. Given the exploratory nature of these analyses, this limitation is acknowledged but was not considered a major concern.

After fitting the LMs and to allow pairwise comparisons among all categories, additional non-parametric analyses were conducted to compare the ECC and EOC across season and Visitor-based seasonality categories. To evaluate differences among all groups, KW tests were performed, followed by Wilcoxon rank-sum post-hoc tests with Bonferroni correction.

Finally, due to the small sample size of F3 and F4, no inferential statistics were applied. Instead, the ECC and EOC from these individuals, collected 50–70 days post-arrival during a potentially stressful period, were compared to those of the main study group using descriptive statistics and graphical representations.

## 3. Results

### 3.1. Assay Validation

For epidermal cortisol, dilution linearity was high (Pearson: r(3) = 0.9880, *p* = 0.007). Mean intra- and inter-assay coefficients of variation were 9.51% and 7.48%, respectively, confirming good repeatability. Spike-and-recovery tests produced an average recovery of 97.94 ± 11.06%, demonstrating assay accuracy. The assay sensitivity (limit of detection) was 25.05 ng cortisol/mL of extract.

For epidermal oxytocin, the dilution linearity test showed an even stronger correlation between expected and measured concentrations (Pearson: r(3) = 0.9994, *p* < 0.001). Mean intra- and inter-assay CVs were 4.48% and 6.00%, respectively. Spike-and-recovery tests yielded an average recovery of 108.34 ± 15.80%. The assay sensitivity was 32.03 pg oxytocin/mL of extract.

### 3.2. Descriptive Overview, Individual Variation, and Correlation of Epidermal Cortisol and Oxytocin Concentrations

Mean ECCs were 4.99 ± 0.88 ng/g (range: 1.57–8.3 ng/g). Although individual maximum levels varied, exceeding 8 ng/g in F2 and M2, and minimum levels dipped below 2 ng/g in M1, no statistically significant differences among individuals were found (KW test: H = 8.21, *p* = 0.084; Figure 1).

Mean EOCs were 44.84 ± 28.81 ng/g (range: 1.46–237.25 ng/g). Mean individual concentrations were generally around 40 ng/g, except for F2, who exhibited a higher mean of 56 ng/g. Maximum values varied widely, ranging from 70 ng/g to over 220 ng/g, while minimum values were more consistent, fluctuating between 1 and 10 ng/g. Statistically significant differences among individuals were found (KW test: H = 11.08, *p* = 0.026; Figure 2). Post-hoc pairwise Wilcoxon rank-sum tests with Bonferroni correction indicated that F2 had significantly higher concentrations than M1 and M3 (*p* = 0.046), while no other pairwise comparisons were significant.

To further contextualize individual differences, the mean ECC and EOC from individuals F3 (N = 3) and F4 (N = 1), collected 50–70 days post-arrival during a potentially stressful context, were compared to those from the main study dataset (F1, F2, M1, M2, and M3; N = 195 samples). The mean ECC of F3 was 6.02 ± 0.88 ng/g, while the single value from F4 was 5.95 ng/g. These corresponded to Z-scores of 1.20 and 1.12 and ranked at the 88.24th and 88.17th percentiles of the dataset for F3 and F4, respectively (Figure 3). Similarly, mean EOCs were also elevated, with F3 showing a mean of 68.43 ± 7.8 ng/g and F4 a single value of 56.82 ng/g. These values corresponded to Z-scores of 0.83 and 0.43 and ranked within the 89.2nd and 73.8th percentiles, respectively (Figure 3).

The Spearman correlation analysis revealed a moderate and statistically significant positive relationship between the ECC and EOC (rho = 0.647, *p* < 0.001; Figure 4).

### 3.3. Descriptive Analysis of Environmental and Welfare-Related Predictors

The selected predictors included welfare-related variables, diazepam administration, and environmental factors that could potentially influence hormone levels. These variables were assessed on a standardized 0–7 scale (i.e., resulting from the sum of daily scores across a week), except for the Negative welfare indicators variable, which was originally summed across multiple individual measures and later normalized for comparability. Table 3 summarizes the descriptive statistics of all predictors considered in the statistical models.

Figure 5 shows no strong correlations among most predictors, apart from diazepam administration with the Closure period (r = 0.57) and Peak season with Summer (r = 0.67). As expected, the seasonal variables (Spring, Summer, and Fall) also showed moderate intercorrelations (~0.35), reflecting natural seasonal overlaps. Diazepam administration increased primarily in 2020, particularly during the Closure period, likely reflecting greater exposure of the ‘subordinate’ dolphins included in the study to social stress, within a unique context of significant husbandry-related challenges such as staff reductions and disrupted schedules due to COVID-19 pandemic restrictions. Although r = 0.57 is near the commonly accepted threshold of 0.6, both variables were retained due to their distinct importance. Similarly, although the correlation between Peak season and Summer exceeded 0.6, both were maintained to differentiate their unique effects: the Peak season relates to visitor attendance, whereas Summer captures environmental and seasonal factors (e.g., water temperature). Nevertheless, these correlations indicate that any findings involving these variables must be interpreted with caution.

The KW test revealed a significant effect of the season on water temperature (χ^2^ = 1261.1, df = 3, *p* < 0.001). Post-hoc pairwise Wilcoxon rank-sum tests with Bonferroni correction showed significant differences between all seasonal comparisons except between Winter and Spring (*p* = 0.26; Table 4).

Finally, Willingness to participate in training sessions was rated ≥3 in 89.29% of 2550 observations (mean = 3.19 ± 0.61; median = 3; mode = 3; range: 1–4), despite missing data on 18.06% of study days.

### 3.4. Effects of Environmental and Welfare-Related Predictors on Epidermal Cortisol Concentrations at Different Time Lags

Different time lags between the predictors and ECC were considered, ranging from 20 to 75 days before sampling, using weekly windows (Table 5). Most predictors did not have a significant effect on the ECC at any time lag, including Negative welfare indicators, Presence of socio-sexual behaviors, Positive social integration, Spring, Peak season, and Closure period.

In contrast, several predictors, including the season, diazepam administration, and Mild weight loss over three months, showed significant relationships with ECC at one or more time lags. However, the specific time intervals at which these associations emerged varied between predictors, with no consistent pattern across all variables (Table 5).

Pairwise analyses for the season revealed a consistent pattern whereby Summer and Fall had notably higher ECCs than both Spring and Winter across most time lags (Appendix A). On average, Fall values were roughly 5–10% greater than Winter and often 10–15% above Spring. Summer consistently exhibited among the highest cortisol concentrations, frequently 5–12% higher than Winter and up to 15% higher than Spring. In contrast, Spring tended to occupy lower values, differing significantly from Summer and sometimes Fall, but showing more modest deviations (5–8%) relative to Winter. Appendix A further illustrates these differences across the eight-time lag windows.

While the LM results did not show a significant effect of the Closure period, the pairwise analyses for Visitor-based seasonality revealed a consistent pattern whereby the Closure period had notably lower ECCs relative to both Off-peak and Peak seasons across most time lags (Appendix A). On average, Closure period values were roughly 5–10% lower than Off-peak and up to 15% lower than Peak season. Meanwhile, the Peak season consistently exhibited the highest ECC, approximately 5–10% above Off-peak season. Off-peak occupied an intermediate range, differing significantly from Peak during certain time lags yet showing only modest deviations (5–7%) from Closure period. Appendix A further illustrates these differences across the eight-time lag windows.

### 3.5. Effects of Environmental and Welfare-Related Predictors on Epidermal Oxytocin Concentrations at Different Time Lags

Different time lags between the predictors and EOC were considered, ranging from 20 to 75 days before sampling, using weekly windows (Table 6). Most predictors did not have a significant effect on ECC at any time lag, including the Presence of socio-sexual behaviors, Positive social integration, Three-month mild weight loss, Spring, Summer, and Peak season. In contrast, several predictors, including Negative welfare indicators, diazepam administration, and season-related variables, showed significant relationships with EOC at one or more time lags. However, the specific time intervals at which these associations emerged varied between predictors, with no consistent pattern across all variables (Table 6).

Pairwise analyses confirmed that Fall showed the highest EOC across most time lags (Appendix A). On average, Fall values were roughly 20–40% above Winter, sometimes reaching 40–50% above Spring. Summer generally followed as the second-highest season, about 10–20% higher than Winter and often 15–30% above Spring. In contrast, Winter and Spring occupied comparatively lower ranges, Winter typically exceeded Spring by 5–10% at shorter lags, though their difference diminished or reversed at later lags. Appendix A illustrates these trends over the eight-time lag windows.

Pairwise analyses for Visitor-based seasonality revealed a consistent pattern whereby the Closure period had a notably lower EOC relative to both Off-peak and Peak seasons, especially from mid to longer time lags (Appendix A). On average, EOCs in the Closure period were roughly 20–30% lower than Off-peak season and up to 40–50% lower than Peak season. Meanwhile, Peak season consistently exhibited the highest oxytocin concentrations, generally 10–25% above Off-peak. The Off-peak season occupied an intermediate range, differing significantly from Peak at certain time lags yet still remaining about 10–20% higher than Closure. Appendix A further illustrates these differences across the eight-time lag windows.

## 4. Discussion

### 4.1. Assay and Method Validation

The proposed sample collection method successfully obtained the epidermis for cortisol and oxytocin analysis in most cases without apparent discomfort, demonstrating its potential as a promising non-invasive, simple, safe, and rapid technique. However, expanding the sampling area beyond 15 cm × 15 cm in future studies may further improve sampling success by increasing the sample mass [27]. Moreover, as detailed by Agustí et al. [27], the amount and appearance of the collected epidermis varied among individuals and seasons. These results are in concordance with Bechshoft et al.’s [30] suggestion of epidermal sloughing occurring in pulses rather than continuously in this species.

Assay validation confirmed that a desquamated epidermis from bottlenose dolphins contains measurable concentrations of cortisol and oxytocin, as demonstrated via our AlphaLISA-based method. These findings are consistent with previous results obtained using Enzyme Immunoassay (EIA) techniques [27,32] and further demonstrate the capacity of the method to provide high linearity, recovery, and precision. Although high-performance liquid chromatography coupled with tandem mass spectrometry (LC–MS/MS) provides greater analytical specificity and sensitivity [30,73], immunoassays, including AlphaLISA, remain more cost-effective and require less specialized equipment or expertise. This affordability and scalability make them especially suitable for wildlife and welfare studies that involve large sample sizes or limited laboratory resources. Cross-reactivity, however, was only evaluated against cortisone (for cortisol) and vasopressin (for oxytocin), and future assessments including additional compounds would be valuable to further confirm assay specificity in cetacean epidermis.

### 4.2. Descriptive Overview and Correlation Between Epidermal Cortisol and Oxytocin Concentrations

The ECC average was 4.99 ± 0.88 ng/g (range: 1.57–8.30 ng/g). When compared to previously reported values, these concentrations aligned with ranges documented for bottlenose dolphins in earlier studies (0.13–8.09 ng/g [27]; and 0.31–16.17 ng/g [30]). The EOC average was 44.84 ± 28.81 ng/g (range: 1.46–237.25 ng/g). These values were notably higher than previously documented concentrations for the stratum corneum in striped dolphins (1.09–9.30 ng/g [32]), though they fell within the wide range measured in deeper epidermal layers of the same species (1.74–337.33 ng/g [32]). However, both similarities and variations observed across studies and species should be interpreted with caution, as absolute concentrations may be affected by analytical methodological differences, including antibody specificity, cross-reactivity, and additional factors inherent to non-immunoassay techniques.

Additionally, notable individual variability was observed in both the ECC and EOC among dolphins. While ECC differences among individuals were not statistically significant, considerable variability was apparent. In contrast, significant differences were detected in the EOC, with dolphin F2 exhibiting significantly higher concentrations compared to dolphins M1 and M3. Such individual differences may be influenced by sex, age, metabolic profiles or individual physiological responsiveness. However, the study design and limited sample size precluded a robust investigation into the specific factors underlying these differences.

The correlation analysis revealed a positive moderate correlation between the ECC and EOC. A similar, though non-significant, trend was reported by Agustí et al. [32] in the stratum corneum of striped dolphins. This relationship aligns with growing evidence that both hormones are co-released during homeostatic challenges and act together to modulate stress responses [40,74]. Although oxytocin is traditionally linked to social bonding and positive welfare in domesticated animals [16], it can also increase under stress. Elevated levels have been observed in humans during socially challenging situations, such as anxiety and interpersonal conflict [75,76], and in animals exposed to physical or environmental stressors [77,78]. These elevations may contribute to stress buffering and adaptation, potentially through inhibitory effects on the HPA axis [77].

### 4.3. Effects of Environmental and Welfare-Related Predictors on Epidermal Cortisol Concentrations

To date, few studies have explored how glucocorticoids and oxytocin are incorporated into the cetacean epidermis or how long they persist after deposition, key aspects that remain largely unknown and are crucial for accurately interpreting our findings. Therefore, we considered a range of plausible time lags (20–75 days) between the activation of the physiological system (central and local HPA axis or oxytocin system) and its detection in the sloughed epidermis, based on current knowledge of hormone biology and the species-specific characteristics of epidermal tissue. Given these uncertainties, we have been cautious in interpreting the associations with environmental and welfare-related variables.

First, we found notable differences in the ECC as a function of the season. Specifically, Summer and Fall consistently exhibited higher ECCs than Winter and Spring. In seasonal species, baseline cortisol concentrations can be very different between seasons [79], potentially driven by environmental factors such as the temperature [80]. In our study, Summer showed the highest water temperatures, with slightly cooler but still warm conditions in Fall. Elevated temperatures may enhance blood perfusion via vasodilation, facilitating heat dissipation and potentially increasing cortisol deposition in the skin, as suggested for blubber in bottlenose dolphins [25,26,80]. Alternatively, these patterns might also reflect seasonal changes in behavior and reproduction, as bottlenose dolphins exhibit more frequent socio-sexual behaviors and elevated reproductive hormones during warmer months [81,82]. Another potential factor could involve seasonal variation in activity, as increased participation in presentations and higher visitor presence may all elevate HPA axis activity [83,84]. However, in this study visitor presence is a less likely contributor as one of the two Summer periods occurred post-COVID-19 with markedly reduced attendance and presentations. Moreover, increased activity during warmer months may increase peripheral perfusion [85], thereby facilitating greater cortisol deposition in the skin.

Neither Peak season nor Closure period showed an effect on the ECC at any of the tested time lags relative to the Off-peak season, although Kruskal–Wallis tests indicated lower ECCs during Closure, intermediate levels in Off-peak, and highest levels in the Peak season. Visitor presence can influence stress responses, but its effects are context-dependent [86,87]. In bottlenose dolphins, some studies report no differences in plasma cortisol between open and closed facilities [88], whereas others suggest higher cortisol during interaction programs with high visitor numbers [89], although uncontrolled seasonal factors may have contributed. Beyond cortisol, visitor presence has also been associated with changes in behavioral diversity [90] and social play [91]. The ECC decrease during the Closure period may thus reflect reduced visitor attendance, but this period also coincided with Spring, whereas the Peak season coincided with Summer, the season most strongly associated with an elevated ECC in the LMs. Additionally, the Closure period overlapped with a higher frequency of diazepam administration, a condition that was associated with a significant decrease in the ECC. Notably, the LMs accounted for these overlapping influences by including the three variables, whereas the KW tests considered only one factor at a time. Therefore, based on these results, visitor presence seems less likely to influence the ECC compared to seasonal or pharmacological effects. However, future studies with more balanced data are necessary to clearly distinguish their respective contributions.

Diazepam administration showed a negative and statistically significant association with the ECC at time lags of 41–54 days. These findings align with the known pharmacological effects of benzodiazepines, which reduce cortisol release following both acute [92,93] and chronic [94] administration in mammals. This effect is primarily mediated by their positive allosteric modulation of gamma-aminobutyric acid (GABA) receptors, which leads to the suppression of the corticotropin-releasing hormone (CRH) system [95,96]. At anxiolytic doses, diazepam is a well-established and safe pharmacological agent used in dolphins to alleviate stress and anxiety during short-term transitional circumstances, such as acclimation to a new habitat or social structure, while also helping maintain appetite and providing mild muscle relaxation [64,68,97].

In the present study, diazepam administration was limited to specific welfare-challenging situations and was primarily directed toward three male individuals characterized as ‘subordinate’ and more susceptible to social conflicts with dominant males. Research indicates that benzodiazepines exert significant behavioral effects in primates, reducing conflict-related behaviors [98,99]. Additionally, diazepam administration has been associated with a reduction in fear-potentiated startle responses and an increase in affiliative behaviors, promoting the development of stable social relationships [100,101]. In the dolphins included in this study, these potential behavioral effects may have also contributed to the reduction in HPA axis activity, not only through the direct pharmacological action of diazepam, but also as a result of reduced social conflict and increased affiliative behaviors.

Weight loss showed a negative and statistically significant association with the ECC at time lags of 55–75 days. This contrasts with previous evidence linking weight loss to elevated cortisol, such as prolonged food restriction in pinnipeds [102,103] and poor body condition in stranded cetaceans [104,105], where fasting may trigger cortisol to help maintain energy homeostasis through lipolysis and gluconeogenesis [106,107]. However, in the present study, weight loss (>5% body mass over three months) represented only a mild to moderate reduction in individuals that generally maintained good body conditions. Therefore, our results may align with findings in humans, where moderate, gradual weight loss caused little to no change [108,109] or even a reduction in plasma cortisol [110].

No significant associations were found between positive social integration or socio-sexual behaviors and ECC at any of the examined time lags. Although changes in HPA axis activity related to social control, affiliative behavior, and socio-sexual interactions have been reported in other mammals [111], several factors may explain our lack of findings. First, the opportunistic methods used to measure these behaviors may have been insufficiently sensitive, producing low or potentially biased frequency data. Second, high inter-individual variability and a small sample size could have reduced the statistical power. Finally, the intensity or duration of these behaviors and its relationships with HPA axis activity may not have been sufficient to induce detectable changes in the ECC, a matrix integrating multiple influences over time that could be overshadowed by stronger factors such as seasonal or pharmacological effects.

Finally, no significant associations were found between Negative welfare indicators and ECC at any of the examined time lags. While our study design differs from previous work, existing evidence supports the use of ECC as a physiological indicator of welfare in cetaceans. For instance, Bechshoft et al. [30] reported elevated ECCs in bottlenose dolphins temporarily removed from the water, while Wong [31] observed similar increases in belugas facing health issues or facility works. In free-ranging stranded striped dolphins, ECCs were positively related to both the severity and duration of welfare compromise [32]. Moreover, studies using other matrices (e.g., feces, blood) have found increased cortisol in bottlenose dolphins in response to negative welfare indicators such as mother–calf separation [112], stereotypic behavior [113], disease [114,115], capture and restraint [116,117,118], and noise exposure [119,120].

The absence of a significant relationship in our study could thus be explained by the low intensity and limited duration of the Negative welfare indicators observed. The mean weekly presence of Negative welfare indicators was minimal, and several critical indicators described in the literature (e.g., poor water quality, prolonged anorexia or apathy, severe aggression [50,54]) were never recorded. In contrast, important positive welfare indicators such as daily food intake and Willingness to participate in training sessions [8,57] remained consistently high, suggesting an overall absence of pronounced welfare challenges. These conditions may not have triggered sufficient activation of the HPA axis, especially considering the limited sample size, to result in detectable increases in the ECC, as has been suggested in other studies for hair cortisol levels [121].

Additionally, in the context of this research, stressors occurring on consecutive days (e.g., certain social conflicts or health issues, which were not frequently observed) were addressed promptly through husbandry or veterinary means, potentially limiting prolonged elevations of circulating cortisol. This interpretation is supported by the distribution of ECCs in our dataset in relation to findings from other studies. Although mean values were higher than baseline levels reported in the same species by Bechshoft et al. [30], they remained well below those associated with assumed stressors. Out-of-water tests in bottlenose dolphins yielded values over ten times higher than baseline [30], while in free-ranging striped dolphins, cortisol levels were up to six times higher in emaciated individuals compared to those in good condition and up to fourteen times higher in those that died from distress-related causes compared to individuals that died from peracute underwater entrapment [32]. By comparison, values in our dataset increased only 1.4 times from the mean to the maximum.

An alternative explanation may lie in limitations in the welfare dataset, which may have reduced the ability to meaningfully link hormonal measures to the animals’ actual welfare state. Although animal caretakers and veterinarians are well positioned to collect welfare-related data [1,5], the absence of systematic observations and a validated scoring system, combined with the opportunistic nature of data collection, may have limited the quality and consistency of the welfare data obtained. Moreover, all negative indicators were treated equally, despite likely differences in their biological relevance in relation to triggering a physiological stress response [122]. Finally, factors such as habituation and potential shifts in HPA axis sensitivity associated with the context of captivity could also be related to these findings [123].

Finally, mean ECC values from F3 and F4 ranked within the upper range of concentrations observed in the main dataset, which could be attributed to enhanced HPA axis activity in response to stressors associated with transport and acclimation. This interpretation is supported by several studies on bottlenose dolphins reporting elevated cortisol in the blood, blubber, epidermis, and feces in response to stressors such as out-of-water restraint [26,30] and interfacility transport [114,124]. Although our sample size for F3 and F4 was small, these findings are in concordance with previous studies suggesting the potential of ECC as a biomarker of HPA axis activity and the welfare status. To build on these insights, we recommend further research leveraging captive cetacean transport, with larger sample sizes and repeated sampling. Such opportunities would be valuable to further validate ECCs as physiological indicators of welfare.

### 4.4. Effects of Environmental and Welfare-Related Predictors on Epidermal Oxytocin Concentrations

In contrast to cortisol, the EOC interpretation remains highly exploratory due to limited prior research. While oxytocin has been primarily investigated in socially relevant contexts [16,125], our study was not specifically designed to capture the fine-scale social dynamics believed to influence its release. Compounding this challenge, the oxytocinergic system is considerably less understood in mammals compared to the HPA axis [16], and in cetaceans, empirical data remain scarce. The interpretation is further hindered by analytical complexities [126] and the unclear relationship between central and peripheral oxytocin concentrations [127,128] and the lack of knowledge regarding how oxytocin may be transported to or incorporated into the skin. As a result, the associations observed between the EOC and welfare indicators in this study, particularly given their occurrence at widely variable time lags, should be interpreted with caution.

First, the EOC displayed a clear seasonal gradient, with Fall consistently yielding the highest values, followed by Summer, and lower concentrations in Winter and Spring. These peaks of the oxytocin concentration coincide with the period when bottlenose dolphins typically show heightened socio-sexual activity and elevated reproductive hormones [81,82], which could therefore align with oxytocin’s role in supporting affiliative and reproductive behavior in mammals [18,24]. However, in light of the correlation observed between oxytocin and cortisol concentrations and oxytocin function in HPA axis regulation (see Section 4.2), and paralleling the seasonal dynamics discussed for the ECC (see Section 4.3), these seasonal patterns may also reflect temperature-driven effects [26,80], as well as seasonal variation in activity, presentation participation, or visitor attendance [83,84].

Visitor-based seasonality analyses showed that the EOC was highest during the Peak season, intermediate in Off-peak periods, and lowest during the COVID-19 Closure, particularly at mid- to long-term lags. This pattern parallels the cortisol response and may arise from similar drivers due to the correlation observed in the two hormones (see Section 4.2), yet its meaning is less clear because oxytocin can increase during both positive social engagement and stress-buffering processes [129]. Therefore, targeted studies tracking socio-sexual and stress-related behaviors more closely, while disentangling these effects from seasonal confound, would be essential to clarify the mechanisms underlying these results.

Diazepam administration was positively associated with the EOC at 20–33-day lags. Because benzodiazepines suppress HPA-axis activity (see Section 4.2), the observed EOC increase might reflect an indirect rebound in affiliative behaviors rather than direct pharmacodynamic stimulation of the oxytocinergic system, an effect that, to date, has not been demonstrated. Interestingly, at the same time lag, a negative association between EOC and Negative welfare indicators was observed. Considering oxytocin’s proposed link to positive welfare states [19,24], higher EOC levels could potentially reflect periods of enhanced welfare, characterized by fewer accumulated negative indicators. Nonetheless, this evidence is far from conclusive, and validating EOC as a positive welfare marker would require consistent covariation with clearly established positive welfare indicators and known contexts.

Overall, although oxytocin has been proposed as a promising physiological indicator of positive welfare, substantial methodological uncertainties and the exploratory nature of current epidermal analyses limit its clear interpretation. A key challenge lies in distinguishing oxytocin’s role as a marker of positive welfare from its broader function in stress regulation, as increases in this hormone may also occur in response to pain or stress [16]. This can be particularly difficult when using integrative matrices such as epidermal tissue, where cumulative hormone deposition may blur distinctions between signals associated with positive affects and those related to stress adaptation. To address these limitations, future research should begin by validating blood or saliva-based oxytocin responses in controlled, socially relevant paradigms capable of detecting subtle welfare-related changes. In parallel, studies elucidating peripheral oxytocin dynamics and the mechanisms underlying its deposition in the skin are essential. These targeted efforts will be critical for advancing the validation of the EOC as a reliable welfare indicator in cetaceans.

### 4.5. Effects of Temporal Dynamics of Cortisol and Oxytocin Incorporation into the Epidermis

A critical step toward establishing the ECC and EOC as valid physiological welfare indicators is to determine the time lags between welfare-related events (i.e., HPA axis or oxytocin system activation) and the subsequent detection of related hormonal concentration changes in the skin. Importantly, these time lags may be different between hormones because cortisol and oxytocin follow distinct endocrine pathways and may be incorporated into the epidermis at different rates. The incorporation of hormones into peripheral tissues depends largely on molecular characteristics such as lipophilicity and the molecular weight. Peptide hormones like oxytocin typically require active transport mechanisms due to their hydrophilic nature and larger size, limiting efficient passage through lipid barriers [35,36]. In contrast, steroid hormones such as cortisol, being small and lipophilic, readily diffuse passively across lipid-rich membranes during keratinocyte differentiation, potentially facilitating their epidermal deposition [35,74].

In the case of cortisol, the proposed mechanism and dynamics of incorporation closely align with models established for other keratinized tissue, such as mammalian hair [130], whale baleen [131], and whale earplugs [28]. According to this model, the deeper (basal) layers of the epidermis reflect more recent hormone exposure, while outer layers, such as the stratum corneum, represent earlier physiological states. This temporal gradient corresponds to the duration of epidermal turnover, which, in bottlenose dolphins, is estimated at approximately 73 days, though it can vary depending on factors such as trauma, hormonal influences, and environmental temperature fluctuations [30,69].

Importantly, the incorporation of hormones from the circulation into peripheral tissues may also depend on peripheral blood flow and the extent of perfusion into the dermis and epidermis, which can determine the degree of exposure to circulating hormones and, consequently, epidermal hormone levels [30]. Peripheral perfusion itself can be influenced by several physiological and environmental factors, including vasodilation and vasoconstriction, acute stress responses, the type and intensity of physical activity, and environmental temperature [25,26,85]. However, the specific conditions and thresholds of perfusion that allow endocrine fluctuations in blood to be reliably reflected in epidermal tissue remain poorly understood and warrant further investigation.

Moreover, interpreting epidermal hormone dynamics is complicated by the fact that cortisol and oxytocin have also been shown to be locally produced within the epidermis in other mammals, where they regulate skin-specific physiological processes such as inflammation, cellular proliferation, oxidative stress responses, and local neuroendocrine signaling [37,132,133]. This local synthesis may independently influence epidermal hormone concentrations, potentially altering the temporal dynamics anticipated solely from systemic hormone secretion. Additionally, epidermal hormone synthesis can stimulate central neuroendocrine responses [134], highlighting the bidirectional connection between local and systemic neuroendocrine responses.

In this study, a key finding that advances our understanding of the time lag between HPA axis activity and cortisol detectability in the sloughed stratum corneum was the observed decrease in cortisol following diazepam administration at time lags between 40 and 54 days. Notably, these intervals closely align with the 43–53-day delay previously reported by Bechshoft et al. [30], further reinforcing the delay reported in that study. Additionally, another key finding supporting it was the relationship of Summer and increased ECC, with significant time lags ranging from 40 to 74 days.

In contrast, the associations between environmental and welfare-related predictors and the EOC emerged at highly variable time lags, complicating a biological interpretation. Notably, interesting relationships with both Negative welfare indicators and diazepam administration were found at shorter time lags (20–26 days). Although these findings are far from conclusive due to limited foundational knowledge about oxytocin dynamics in cetaceans, they highlight an important consideration: differences in molecular properties and transport mechanisms between cortisol and oxytocin could significantly affect the interpretation of their epidermal concentrations in relation to welfare states.

Another critical step in validating a new matrix is to determine the time window over which hormones are integrated into it. The length of this integration window likely depends on factors such as the epidermis’ growth rate and the depth of the sampled tissue. In parallel, the duration and magnitude of a stressor, and therefore of hormone exposure, may determine the ability to detect hormonal changes using the epidermis [44]. Thus, beyond understanding the integration window, it is essential to understand the sensitivity of this biological matrix to adrenocortical responses across a range of intensities (from mild to severe) and durations (from brief to prolonged) [135,136]. Building on these considerations, full-depth epidermal samples likely represent hormone levels throughout the skin renewal cycle, offering a broader picture of the physiological state over time [32,73]. This integration may mask brief or minor hormone fluctuations, reducing sensitivity to acute states [44], but enhance the detection of prolonged circulating hormonal changes [135]. In contrast, the superficial layers of the stratum corneum may represent a narrower temporal window of hormonal activity and could be useful for identifying acute, high-intensity changes as suggested by Bechshoft et al. [30]. In our study, although only discrete 7-day intervals were analyzed, significant declines in ECCs across adjacent blocks (e.g., 41–47, 48–54 days) could indicate a roughly two-week window of hormonal integration.

In the case of this study, the observed relationships between the ECC and certain conditions, such as pharmacological treatments, extended seasonal fluctuations in HPA axis activity, or inter-facility transport and acclimatization processes, could therefore be related to their greater intensity or longer duration. Conversely, the lack of associations with Negative welfare indicators might reflect the sporadic or lower-intensity nature of these events, as well as their prompt resolution through husbandry or veterinary interventions, potentially limiting prolonged elevations in circulating cortisol. However, since the specific intensity and duration of each condition were not explicitly evaluated in this study, further research is necessary to clarify these factors.

### 4.6. Study Limitations and Future Directions

This study faced several limitations that should be considered when interpreting the findings. Most notably, the relatively small sample size constrains the generalizability of the results and may have limited the statistical power to detect more subtle associations. The reliance on welfare data derived from routine monitoring, while valuable, may have lacked the resolution necessary to capture fine-scale changes, particularly given the overall stability in the animals’ welfare status during the study period. Relatedly, the absence of an overall welfare score or a more detailed characterization of welfare states and the social context further limited the ability to provide valuable context for interpreting ECC and EOC results. Furthermore, the current lack of detailed knowledge regarding hormone dynamics in the epidermis restricts the ability to fully interpret the ECC and EOC.

These limitations highlight the need for future research involving larger, more diverse populations and more detailed, standardized welfare assessments. Additionally, further research is essential to better understand the temporal dynamics of cortisol and oxytocin incorporation into the epidermis, with particular attention to how peripheral perfusion influences the extent and timing with which changes in circulating concentrations are reflected in epidermal tissue. Experimental approaches such as adrenocorticotropic hormone (ACTH) challenges in captive animals [137,138] and the use of radioisotope-labeled hormones [139], ideally taking seasonality and temperature variability into account, could help elucidate the time lag, integration window, and sensitivity of each hormone incorporation and deposition in the skin. In addition, opportunistic studies involving a broader range of stressors, varying in nature, duration, and intensity, would be highly valuable for establishing detection thresholds and enhancing the interpretability of epidermal hormone data in the context of cetacean welfare assessments.

## 5. Conclusions

This study successfully validated the analytical measurement of the ECC and EOC in bottlenose dolphins using AlphaLISA assays, while also providing meaningful results that contribute to their ongoing biological validation as physiological welfare indicators in cetaceans. When considered alongside previous findings, the ECC appears to be a valid biomarker for assessing retrospective, intermediate-term welfare changes in living cetaceans. However, in practice, it should ideally be integrated into a broader, multifactorial welfare framework that incorporates complementary physiological and behavioral measures. The ECC could also represent a practical and scalable welfare indicator for both managed and free-ranging cetaceans, offering key advantages such as ease of sampling, minimal disturbance, and adaptability across diverse contexts. As such, the ECC holds considerable potential for advancing our understanding of the chronic and cumulative impacts of multiple stressors, particularly in endangered species where non-invasive monitoring tools are critically needed. By contrast, the EOC, while offering some relevant insights, appears less reliable at this stage of knowledge, partly due to the limited understanding of the oxytocinergic system and its dual role in both positive and negative affective states, which complicates its current applicability as a welfare indicator.

## Figures and Tables

**Figure 1 animals-15-02628-f001:**
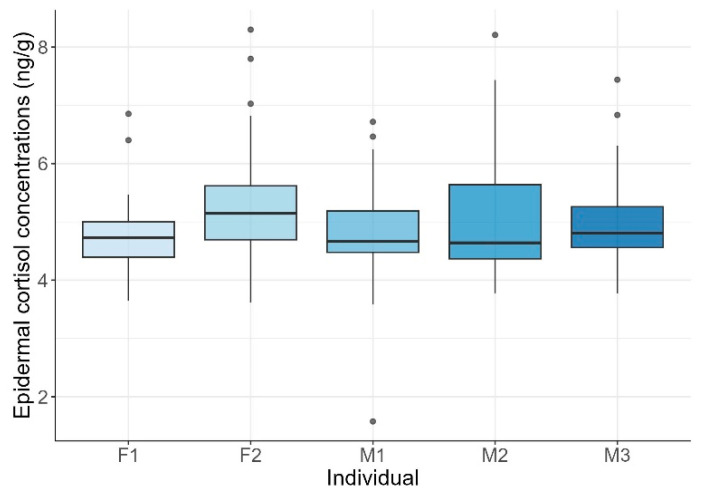
Epidermal cortisol concentrations across common bottlenose dolphin (*Tursiops truncatus*) individuals. Boxplots show the distribution of cortisol concentrations for each female (F1, F2) and male (M1, M2, M3) dolphin. Horizontal lines represent the lower quartile, median, and upper quartile values, while whiskers indicate the range. Points represent outlier observations.

**Figure 2 animals-15-02628-f002:**
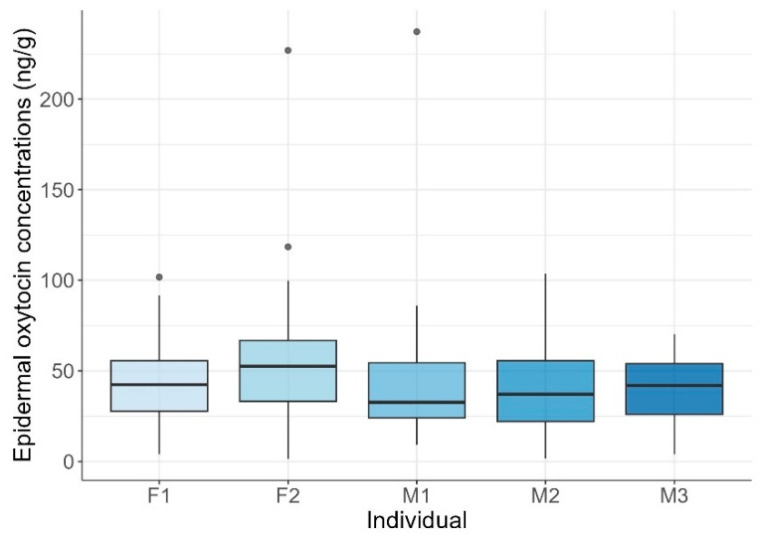
Epidermal oxytocin concentrations across common bottlenose dolphin (*Tursiops truncatus*) individuals. Boxplots show the distribution of oxytocin concentrations for each female (F1, F2) and male (M1, M2, M3) dolphin. Horizontal lines represent the lower quartile, median, and upper quartile values, while whiskers indicate the range. Points represent outlier observations.

**Figure 3 animals-15-02628-f003:**
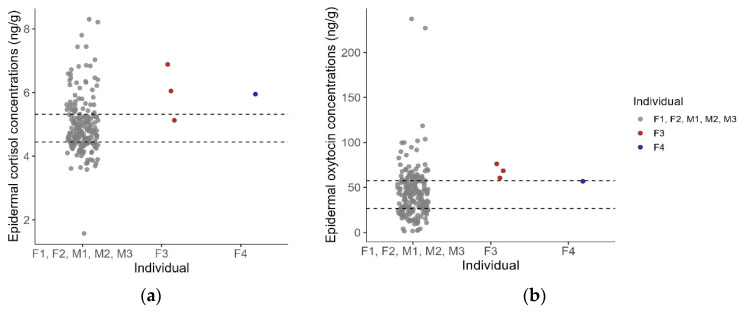
(**a**) Epidermal cortisol concentrations across three common bottlenose dolphin (*Tursiops truncatus*) groups. The plot displays individual data points grouped into three categories on the x-axis: main individuals (F1, F2, M1, M2, M3), individual F3, and individual F4. Each dot represents a single epidermal sample. Horizontal lines indicate the 25th and 75th percentiles of the full dataset (N = 195), providing a reference for the interquartile range; (**b**) epidermal oxytocin concentrations shown using the same grouping and visual format.

**Figure 4 animals-15-02628-f004:**
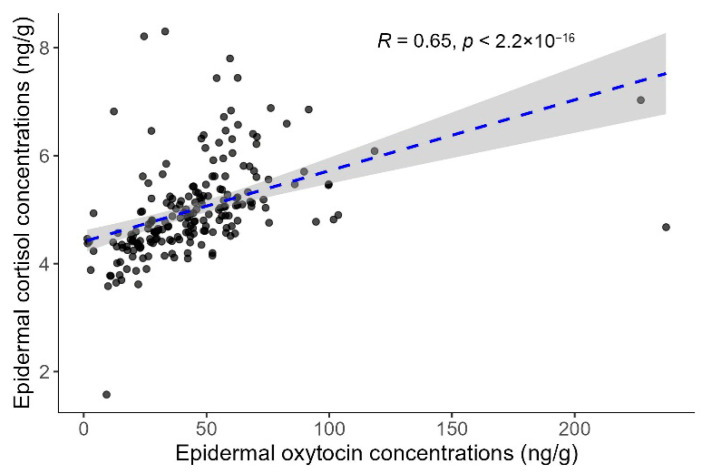
Correlation between epidermal cortisol and oxytocin concentrations in common bottlenose dolphins (*Tursiops truncatus*). Each point represents an individual sample. The dashed blue line indicates the linear regression fit, and the grey shading represents the 95% confidence interval around the regression line.

**Figure 5 animals-15-02628-f005:**
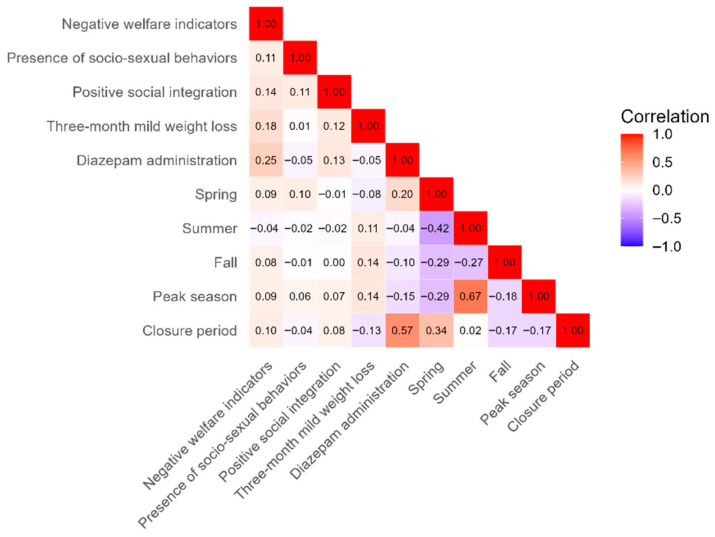
Correlation matrix of predictor variables used in statistical models relating to environmental and welfare-related conditions in common bottlenose dolphins (*Tursiops truncatus*). Pairwise correlations were calculated using Spearman’s rank correlation coefficient (ρ).

**Table 1 animals-15-02628-t001:** Common bottlenose dolphin (*Tursiops truncatus*) welfare indicators identified in aquarium daily reports, classified according to the Five Domains Model [42]. For each indicator, the corresponding domain (Domains 1–4), a brief description, and its potential contribution to the animal’s affective experience (Domain 5) are provided, along with supporting references for their interpretation.

Domain	Welfare Indicator	Description	Type Contribution to Welfare State (Positive/Negative) and Inferred Affective States (Domain 5)	References
**Domain 1:** **Nutrition**	Three-month mild weight loss	Calculation of the Body Weight Oscillation Score (BWOS) was performed by determining the difference between the maximum and minimum body weight over a three-month period, dividing this value by the average body weight, and multiplying by 100. A BWOS exceeding 5% was considered indicative of mild weight loss. In our results, BWOS ranged from 5% to 13.5%, with a maximum three-month weight loss of 13.5% of average body weight. Dolphins were voluntarily weighed approximately once per month using a scale located outside the pool. To obtain daily body weight values for calculating BWOS, weights were interpolated between consecutive measurements, assuming a gradual and continuous weight change throughout each interval.	Negative; long-term hunger, weakness	[54]
	Reduced food intake	Measure of the percentage of fish consumed daily by each animal, defined as days when intake was below 90% of the total food offered (recorded in kg).	Negative; short-term hunger, malaise	[8,57,58]
**Domain 2:** **Environment**	Social isolation	Observations of dolphins kept alone and separated from their social group for extended periods, resulting in impeded affiliative interactions and reduced opportunities for social behaviors.	Negative; loneliness, insecurity, anxiety	[58,59]
**Domain 3:** **Health**	Incidence of eye lesions	Observations of eye opacities, corneal scars, color changes, or any other indicators of eye lesions or diseases.	Negative; malaise, pain, sickness, exhaustion, and, where applicable, breathlessness	[53,60]
	Incidence of oral and dental conditions	Observations of dental wear, fractures, missing teeth, gingivitis, tongue injuries, fungal lesions, or mucosal lesions, including records of previous oral diseases.		[53,61]
	Incidence of respiratory diseases	Observations of blowhole secretions, odors, unusual sounds, or changes in respiratory rate, along with diagnosis of respiratory infections or diseases.		[53,62]
	Incidence of gastrointestinal diseases	Observations of signs indicating gastrointestinal dysfunction, including records of gastric and fecal abnormalities, cytological evaluations, cultures, and parasitological examinations.		[15,63]
	Incidence of renal conditions	Observations of renal abnormalities, including blood chemistry and urinalysis results.		[53,64]
	Incidence of skin lesions and diseases	Observations of viral, fungal, or bacterial skin lesions, wounds, discoloration, or other abnormalities.		[53,64]
**Domain 4:** **Behavioral** **Interactions**	Negative social integration	Observations of limited or absent engagement with a novel social group, or the emergence of agonistic interactions after attempted integration.	Negative; anger, anxiety, fear, insecurity	[53,58]
	Presence of mild or low-intensity agonistic behaviors	Observations of agonistic social events or states (e.g., chasing, biting, jaw clapping), including the presence of new rake marks.		[53,58]
	Regurgitation	Observations of individuals ejecting a full fish or a mixture of water and fish remnants, often followed by re-swallowing the same material.	Negative; boredom, depression, anxiety	[54]
	Foreign body ingestion	Observations of individuals ingesting non-food objects, or documentation of the item’s removal.		[59]
	Presence of socio-sexual behaviors	Observations of dolphins engaging in socio-sexual behaviors, including genital inspection or contact, or movements in close genital proximity.	Positive; affectionate sociability, excitation/playfulness, sexually gratified	[53,54]
	Positive social integration	Observations of affiliative social events or states (e.g., gentle approaches, synchronous swimming, playful contact) following integration with a novel social group.		[7,65]
**Domain 5: Mental state**	Willingness to participate in training sessions (positive human–animal relationship)	Daily records of willingness to participate in sessions as rated on a 5-point scale (0 to 4) representing incremental dolphin’s motivation and enthusiasm during training sessions.	Low: anxiety, fear, insecurity, non-compliant, avoidance High: calm, confident, feels in control, compliantly responsive, seeks contact, bonded with humans	[8,66,67]

**Table 2 animals-15-02628-t002:** Overview of predictor variables used in statistical models assessing the relationship between epidermal hormone concentrations and environmental and welfare-related conditions in common bottlenose dolphins (*Tursiops truncatus*). Indicator definitions are provided in Table 1.

Predictor	Description
Negative welfare indicators	Weekly sum of daily scores of the Negative welfare indicators variable, which includes the cumulative daily presence of the following welfare indicators: Reduced food intake, Social isolation, Incidence of eye lesions, Incidence of oral and dental conditions, Incidence of respiratory diseases, Incidence of gastrointestinal diseases, Incidence of renal conditions, Incidence of skin lesions and diseases, Negative social integration, Presence of mild or low-intensity agonistic behaviors, Regurgitation, and Foreign body ingestion
Presence of socio-sexual behaviors	Weekly sum of daily scores of Presence of socio-sexual behaviors
Positive social integration	Weekly sum of daily scores of Positive social interactions
Three-month mild weight loss	Weekly sum of daily scores of Weight loss over a three-month period
Diazepam administration	Weekly sum of daily scores of diazepam administration
Spring	Weeks containing within 21 March–20 June
Summer	Weeks within 21 June–20 September
Fall	Weeks within 21 September–20 December
Peak season	Weeks within 1 July–31 August 2019
Closure period	Weeks within 12 March–30 June 2020

**Table 3 animals-15-02628-t003:** Descriptive statistics of predictor variables used in statistical models relating epidermal hormone concentrations to environmental and welfare-related conditions in common bottlenose dolphins (*Tursiops truncatus*).

Predictor	Mean (Scale 0–7)	SD	Median	Min.	Max.
Negative welfare indicators (normalized) *	0.19	0.21	0.142	0	1.214
Presence of socio-sexual behaviors	0.15	0.48	0	0	3
Positive social integration	0.43	0.91	0	0	6
Three-month mild weight loss	0.34	1.43	0	0	7
Diazepam administration	1.60	2.87	0	0	7
Spring	1.91	3.12	0	0	7
Summer	1.59	2.94	0	0	7
Fall	1.58	2.93	0	0	7
Peak season	0.65	2.02	0	0	7
Closure period	1.08	2.48	0	0	7

* All welfare indicators negatively affecting welfare were summed into the Negative welfare indicators variable. With 12 indicators, its original range is 0–84. To align with the scale of other predictors (0–7), this variable was normalized by dividing by 12.

**Table 4 animals-15-02628-t004:** Seasonal variation in common bottlenose dolphin (*Tursiops truncatus*) facility water temperature, with post-hoc pairwise Wilcoxon comparisons following a Kruskal–Wallis test.

Season	Mean ± SD (°C)	Winter (*p* Value)	Spring (*p* Value)	Summer (*p* Value)	Fall (*p* Value)
Winter	20.8 ± 0.89	-	0.26	<0.001	<0.001
Spring	21.4 ± 1.93	0.26	-	<0.001	<0.001
Summer	23.2 ± 1.09	<0.001	<0.001	-	<0.001
Fall	22.6 ± 0.70	<0.001	<0.001	<0.001	-

**Table 5 animals-15-02628-t005:** Predictor estimates from linear models (LMs) for epidermal cortisol concentrations in common bottlenose dolphins (*Tursiops truncatus*) across different time lags (in days). Green cells represent statistically significant (*p* < 0.05) positive relationships, while red cells indicate statistically significant (*p* < 0.05) negative relationships. Yellow cells indicate trends with *p*-values between 0.05 and 0.1 (0.1 > *p* > 0.05).

Predictors	Time Lag in Days (Estimates)							
	20–26	27–33	34–40	41–47	48–54	55–61	62–68	69–75
Negative Welfare Indicators	0.014	−0.017	−0.025	0.028	−0.037	−0.08	−0.042	−0.015
Presence of socio-sexual behaviors	−0.026	0.039	0.015	−0.029	−0.028	0.049	0.023	0.040
Positive social integration	0.129.	0.083	0.035	−0.044	0.085	−0.003	0.017	0.035
Three-month mild weight loss	−0.050	−0.038	0.003	−0.033	−0.095	−0.179 **	−0.224 ***	−0.206 **
Diazepam administration	−0.108	−0.148.	−0.136.	−0.178 *	−0.229 **	−0.132.	−0.126.	−0.116
Spring	−0.065	−0.091	−0.079	−0.053	−0.052	−0.045	0.035	0.067
Summer	0.1563	0.194.	0.174.	0.233 *	0.158	0.204 *	0.316 **	0.322 ***
Fall	0.170 *	0.142.	0.146.	0.136.	0.119	0.108	0.076	0.095
Peak season	−0.013	−0.083	−0.035	−0.065	0.048	0.007	−0.027	−0.070
Closure period	−0.066	−0.001	0.024	0.048	0.120	0.038	−0.022	0.014

***: *p* < 0.001; **: *p* < 0.01; *: *p* < 0.05; .: *p* < 0.1.

**Table 6 animals-15-02628-t006:** Predictor estimates from linear models (LMs) for epidermal oxytocin concentrations in common bottlenose dolphins (*Tursiops truncatus*) across different time lags. Yellow cells indicate trends with *p*-values between 0.05 and 0.1 (*p* > 0.05 < 0.1). Green cells represent statistically significant (*p* < 0.05) positive relationships, while red cells indicate statistically significant (*p* < 0.05) negative relationships.

Predictors	Time Lag (Estimates)							
	20–26	27–33	34–40	41–47	48–54	55–61	62–68	69–75
Negative welfare indicators	−5.140 *	−2.353	−3.195	−1.373	−0.912	−0.319	−2.433	−1.813
Presence of socio-sexual behaviors	2.139	−0.232	−1.55	−3.579	−2.214	−0.463	3.135	−2.434
Positive social integration	−0.494	1.040	−0.197	−2.949	−0.368	1.763	2.85	−1.155
Three-month mild weight loss	0.753	−0.855	−1.419	−1.252	−0.362	−3.126	−3.883.	−3.761.
Diazepam administration	5.702 *	4.673.	3.924	3.951	0.986	0.550	0.155	0.813
Spring	−2.577	−2.244	0.669	1.812	−0.159	0.189	1.688	4.921.
Summer	−1.054	−0.565	−1.759	−0.712	−1.377	1.167	5.095	5.606.
Fall	5.133 *	4.607.	5.696 *	6.069 *	4.614.	4.893.	4.372.	5.903 *
Peak season	4.918.	4.469	6.468 *	6.508 *	5.923.	4.441	0.094	−0.030
Closure period	−2.959	−3.848	−4.678	−5.235.	−3.719	−5.059.	−5.502 *	−6.745 **

**: *p* < 0.01; *: *p* < 0.05; .: *p* < 0.1.

## Data Availability

The raw data supporting the conclusions of this article will be made available by the authors on request.

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
