# Peer review of "Tracking Epidermal Cortisol and Oxytocin in Managed Bottlenose Dolphins as Potential Non-Invasive Physiological Welfare Indicators"

_animals, 2025, doi:10.3390/ani15172628_

Round 1
Reviewer 1 Report
Comments and Suggestions for Authors
Agusti and colleagues have presented a study of potential endocrine markers of welfare in managed-care bottlenose dolphins. The topic is certainly of interest given public concern over the welfare of these large-brained mammals under human care. The study assesses the potential of using the endocrine profile of sloughed skin to monitor historical indicators of welfare; cortisol is used as the classic marker of the stress response, and oxytocin is used a novel marker, but about which much less is known. I think the paper is well-written and I have only a handful of comments that I ask the reviewers to consider. With minor modification, I think this article is suitable for publication.
The overall study design could have been improved if a known stressor or increase in hormone (e.g. oral hydrocortisone) was applied at a specific date and the subsequent skin analysis benchmarked against that event. However, I do not expect the authors to repeat the study. What they have done has value.
One area that should be discussed further is the issue of peripheral vasodilation and the need to deliver hormone to the periphery of the blubber. This is mentioned in lines 679-683, but it is likely the critical factor determining whether a strong signal occurs in the skin. Is peripheral blood flow sufficient to deliver the cortisol or oxytocin such that epidermal cells can uptake the hormone? This phenomenon is apparent when comparing Champagne et al. 2018 with Bechshoft et al. 2020. The dolphin stress test of Champagne et al. provided the significant cortisol increase used as the cortisol signal in Bechshoft et al. It is not surprising that the animal (COL) with the greatest accumulation of cortisol in the blubber (Champagne et al), also demonstrated the strongest signal in the skin (Bechshoft et al). Similarly, the dolphin BLU was not included in the blubber analysis of Champagne et al. because the blubber levels were already very high prior to the stress test (thus providing no baseline with which to compare). BLU was the other animal in which the skin cortisol signal was readily seen. Results were less clear for the dolphins with lower blubber cortisol levels. The impact of blubber perfusion should be further discussed as lack of significant peripheral diffusion is likely a major factor in how effective this approach will be as a monitoring tool.
As an editorial observation, the Discussion is quite long. I think the authors should consider how they might make it more concise. However, this is my opinion, and I leave it to the authors to determine whether they agree.
Lines 73-75: There can be many disruptions of homeostasis that do not trigger the HPA axis. This statement is too broad and fails to highlight a primary role in addressing acute or chronic stress, which is your most direct link to animal welfare. Please consider alternative wording. Also, the reference provided addresses the effect of bilateral adrenalectomy on ACTH production. It is very specific, and seems an odd and not very supportive reference given the statement it follows. I suggest finding a more general article about the HPA axis to support your refined statement, such as:
Reeder, D. M. and K. M. Kramer (2005). "Stress in free-ranging mammals: Integrating physiology, ecology and natural history." Journal of Mammalogy 86(2): 225-235.
Line 150: Outside of the younger males for which an explanation of potential welfare issues were stated, what made the other animals more likely to exhibit welfare issues?
Line 161: I am not familiar with the phrase “thermal comfort zone,” nor am I sure how one would go about measuring it. If you mean “thermal neutral zone,” then I concur. There are empirical measurements to support the temperatures being thermally neutral.
Line 176: Here it states that two epidermal samples were collected per individual, but the prior sentence lists one animal three times and the other only once. Please clarify as the two sentences do not appear to agree with each other.
Line 182: I am curious as to why the dorsal fin was used and not the lateral or dorsolateral trunk. Can you provide your rationale for selecting this site?
Lines 207-209: Are there other compounds that have greater cross-reactivity? Typically, there are myriad compounds with varying levels of cross-reactivity. If there are compounds with higher cross-reactivity than reported for cortisone or oxytocin, please include them for the sake of completeness.
Line 300: I believe analysis should be analyses.
Figures 1 and 2: It would be better to state that the points represent outlier observations, which is typical of a box-and-whisker plot.
Line 430: Post-hoc, not Post hoc.
Lines 563-572: This should all be one paragraph.
Lines 638-644: You should be careful to compare absolute measures among different techniques, particularly those involving different antibodies. Patterns are more useful for comparison across studies as the binding affinity and cross-reactivities will affect absolute concentrations and will vary as a function of the antibody, whereas non-immunoassays will have other factors affecting reported concentrations. You mention this in line 646, but it comes across as a passing thought.
Line 707-709: I would be cautious with this statement as I do not believe Matsushiro et al. 2021 accounted for seasonal differences in cortisol. This is a confound in interpreting their results.
Lines 841-848: Is this an opportunity to link how water temperature may be related to a greater degree of regular peripheral vasodilation?
Line 923: Please add duration and magnitude of exposure to the hormone to “growth rate and the depth of the sampled tissue.”
This review prepared by Dorian S. Houser.
Reviewer 2 Report
Comments and Suggestions for Authors
Thank you for this comprehensive study. Overall this is a strong paper with robust methods, and the authors do not overstep the conclusions they draw, but rather make appropriate suggestions about connections and future research steps. Given its length, and the fact that several sentences are stated twice in separate areas, the major suggestion that I would have would be to review the paper for conciseness and clarity. Minor comments are below.
Line 92, 133, etc. : in some places 'bottlenose dolphin' is used, while in others 'common bottlenose dolphin' is used. Suggest uniform use throughout, perhaps first listing as common bottlenose dolphin with species. (Additionally, the scientific name is listed in line 123 and again in line 146; consider omitting second.)
Line 116: suggest listing in the text what the "e.g. [45] [46]" is referencing (beluga whales).
Line 137-140, line 165-167: this information is very similar/repeated in the ethics section and individuals section; consider either consolidating or reframing.
Line 161: consider a reference for this species' thermal comfort zone.
Line 179: consider simplifying by omitting 'in a 'line up' procedure' (which is likely a facility-specific term) and simply describing it with the rest of the sentence.
Section 2.5: consider describing whether one or more people were involved in the sample detection/measurements, as this could affect accuracy as well?
Table 1: for column, "Type contribution to welfare state (posi-tive/negative) and in-ferred affective states (Domain 5)", Domain 3 (Health) is made up of multiple rows, and it is confusing the way it is currently displayed, suggesting that the incidence of eye lesions is associated with breathlessness? I believe this may be simply a question of formatting.
Line 276: "Significant differences were found" consider moving to results.
Line 291-293, and line 348-350: again, these are almost identical sentences; based on the length of the article consider removing one of these to reduce redundancies.
Table 2: Consider removing each instance of "see table 1 for indicator description" and moving a general comment to that effect to the Table 2 legend.
General comment for lines 314-324, and Table 2 predictor descriptions: while I understand what you are trying to convey with the reason for aggregating weekly spans, this paragraph and the descriptions of each of the seasonal predictors reads as unnecessarily complex. Consider clarifying/omitting some of this to what is truly vital to convey the aggregation, and consider rewording the descriptions like 'weeks containing or fully within'.
Lines 336-338 and lines lines 361-363: the time lags are repeated here. Consider removing and replacing with something like 'tested for the previously described time lags'.
Line 415-416: looking at the graph, this sentence does not appear to match: "Although individual maxi-415 mum levels varied, exceeding 8 ng/g in F2 and M2 and dipping below 2 ng/g in M1": dipping below 2 ng/g does not appear to be a maximum for this individual, but rather a minimum?
Tables, general comment: the supplementary files make sense for supplementary information; consider whether all of the tables used are critical in the body of the text or if any could be moved to supplementary files as well.
Line 622-624, and line 894-896: you mentioned the average epidermal turnover of 73 days, but in the lines above described pulsed sloughing. Do you discuss the influence that this pulsed sloughing may have had on lag time windows (e.g. is there a change with seasonality on how quickly the epidermis is sloughed and so does that change the window that it corresponds to?)
